# One-Step Coating Processed Phototransistors Enabled by Phase Separation of Semiconductor and Dielectric Blend Film

**DOI:** 10.3390/mi10110716

**Published:** 2019-10-24

**Authors:** Lin Gao, Sihui Hou, Zijun Wang, Zhan Gao, Xinge Yu, Junsheng Yu

**Affiliations:** 1State Key Laboratory of Electronic Thin Films and Integrated Devices, School of Optoelectronic Science and Engineering, University of Electronic Science and Technology of China (UESTC), Chengdu 610054, China; lin_gaoedu@163.com (L.G.); shhou@std.uestc.edu.cn (S.H.); wangzijun@std.uestc.edu.cn (Z.W.); 2Department of Biomedical Engineering, City University of Hong Kong, Hong Kong 999077, China; zhangao2-c@my.cityu.edu.hk

**Keywords:** organic thin-film transistors (OTFTs), phototransistor, vertical phase separation, polymer blend

## Abstract

Fabrication of organic thin-film transistors (OTFTs) via high throughput solution process routes have attracted extensive attention. Herein, we report a simple one-step coating method for vertical phase separation of the poly(3-hexylthiophene-2,5-diyl) (P3HT) and poly(methyl methacrylate) (PMMA) blends as semiconducting and dielectric layers in OTFTs. These OTFTs can be used as phototransistors for ultraviolet (UV) light detection, where the phototransistors exhibited great photosensitivity of 597.6 mA/W and detectivity of 4.25 × 10^10^ Jones under 1 mW/cm^2^ UV light intensity. Studies of the electrical properties in these phototransistors suggested that optimized P3HT contents in the blend film can facilitate the improvement of film morphology, and therefore form optimized vertical phase separation of the PMMA and P3HT. These results indicate that the simple one-step fabrication method creates possibilities for realizing high throughput phototransistors with great photosensitivity.

## 1. Introduction

Photodetectors have attracted extensive attention due to their potential applications in intelligent fields such as sensors, optical communications, military security, displays, imaging, and biomedical instruments [1,2,3,4,5,6,7]. Among them are organic phototransistors that use organic thin film transistors (OTFTs) with light sensitive semiconductors as the basic platforms for photo detection [8,9,10]. Compared to conventional photodiodes, phototransistors are controllable devices whose carrier density in the channel region can be tuned by the gate electrode, and therefore act as an in-built amplifier for enhancing the signal to noise ratio [2,11,12]. In recent years, investigations of phototransistors have mainly focused on the device structure design, fabrication process optimization, and material synthesis and selection, where the basic OTFT platforms are based on multiple layer structures [13,14,15,16]. The functional layers in the OTFTs, including electrodes, the dielectric layer, and semiconducting layers, are typically processed by thin film deposition techniques with a stacked layer-by-layer geometry [1,9]. Solution processes such as spin-coating are common methods for fabricating phototransistors; however, challenges exist for multistep solution processing due to the swelling and dissolution of the underlying film during spin coating of the upper layer [17,18]. Although the use of orthogonal solvent is a feasible way to avoid this problem, it limits the selection of materials; some defects would inevitably be introduced into the interface, and it is hard to control the crystallinity and morphology of the dielectric film [19,20]. Therefore, it is important to find a simple and effective method to break the limitations of the above problems.

Recently, efforts towards high performance OTFTs have been reported using organic semiconducting (OSC) materials and insulating polymer blends [21,22]. For example, the polymer/polymer blend of diketopyrrolopyrrole–thiophene (DPP–T; P1) and poly(N-vinyl carbazole) (PVK) exhibited great semiconducting properties with stable charge transport at high temperatures [23]; the small molecule/polymer blend of 2,7-dioctyl(1)benzothieno(3,2-b)(1)benzothiophene (C8-BTBT) and polystyrene (PS) afforded a high field-effect mobility exceeding 10.0 (cm^2^/Vs) [24]. Furthermore, for regulation of the semiconductors’ crystallinity during the phase separation process, ultra-thin flexible OTFT devices and high-performance gas sensors were realized by poly(3-hexylthiophene-2,5-diyl) (P3HT):poly(methyl methacrylate) (PMMA) blend films [25,26,27,28]. These existing works have also indicated that ideal vertical phase separation between insulating polymers and organic semiconductors may happen during the coating process and thus provide a new simple route for a one step process for both dielectrics and semiconductors. Among the OSC:polymer blend systems, P3HT:PMMA blend film with a certain proportion can form a self-stratified structure with a PMMA-bottom and P3HT-top by one-step spin coating [29]. In addition, owing to the ability of capturing and converting incident light into detectable electrical signals, P3HT was also widely used as organic semiconductor material of detectors [30,31,32]. Excitingly, combining the advantages of P3HT in phase separation in the blend system and its unique optical absorption capability would be a very good direction to realize high sensitivity phototransistors with simple fabrication processes.

In this paper, we reported a simple one step spin-coating route for realizing phototransistors based on vertical phase separation, which associates with self-stratified semiconductor/dielectric bilayer structures. The band gap of a P3HT semiconductor decides the photo-generated charge carriers, while the intrinsic mobility promotes the transition in the conductive channel from photo-generated and gate-modulated carriers to an electrical signal. Firstly, the interface of air/film was analyzed to ensure the vertical phase separation happened. In addition, relative parameters of phototransistors were tested and discussed. Beyond that, we investigated the influence of P3HT contents on the phototransistor performance and explored the underlying mechanism, which contributes to developing high performance phototransistors via vertical phase separation.

## 2. Experimental

### 2.1. Materials Preparation

P3HT (*M*_W_ = 10–100 k) was purchased from Xi’an Polymer Light Technology Corp. (Xi’an, China), PMMA (*M*_W_ = 120 k) was purchased from Sigma-Aldrich (St. Louis, MO, USA), and chlorobenzene was purchased from Tokyo Chemical Industry Company (Tokyo, Japan). To prepare blend materials for vertical phase separation, pure conjugated polymer P3HT and PMMA were dissolved in chlorobenzene solvent, and mixed at the ultimate concentration of P3HT (1.0, 2.0, and 5.0 mg/mL) and PMMA (10 wt%). The solutions were stirred on a magnetic stirring plate overnight to ensure sufficient dissolution and mixing; the rotational speed and temperature were controlled at 300 rpm and 80 °C, respectively. All the materials were used as received without further purification.

### 2.2. Device Fabrication

All devices with a bottom-gate and top-contact structure were fabricated on indium tin oxide (ITO) glass substrates, which were cleaned by sonication in chloroform, acetone and isopropyl alcohol for 15 min each step. Pre-cleaned ITO was treated with ultraviolet (UV)-zone for 15 min after being dried in an oven beforehand. A P3HT-top and PMMA-bottom bilayer structure was formed in the process of one-step spin coating at 1500 rpm for 30 s in a glovebox, followed by annealing at 60 °C on a hot plate for 1 hour to remove residual solvents. Gold source-drain electrodes were evaporated about 50 nm through a shadow mask with channel width (W) and length (L) of 100 μm and 1 cm, respectively.

### 2.3. Measurement and Characterization

All the electrical measurement in dark and under illumination were characterized by using a Keithley-4200 semiconductor parameter analyzer (Tektronix, Inc., Beaverton, OR, USA) under ambient conditions. Noise current was obtained by putting OTFTs in a black box, and the photocurrent was obtained under the excitation of a simulated solar spectral light source with a 365 nm bandpass filter. The absorption of the spectrum and the inner molecular packing structure was further analyzed by UV–vis spectrophotometer (SHIMADZU UV-1700) measurements (Shimadzu Corporation, Kyoto, Japan). A non-contact surface profiler (Contour GK-T) (Bruker, Karlsruhe, Germany)) was used to obtain the thickness of the films. Atom force microscopy (AFM) (MFP-3D-BIO, Asylum Research) (Oxford Instrument, Oxford, UK)) in a tapping mode was used to observe the surface morphologies. The phase separation of blend films was evaluated by transmission electron microscopy (TEM, FEI Philips Tecnai 12 BioTWIN) (FEI, Hillsboro, OR, USA) at an accelerating voltage of 200 kV.

## 3. Results and Discussion

### 3.1. Thin Film Characterization of Devices

Figure 1a illustrates a schematic diagram of a phototransistor consisting of P3HT and PMMA blend films on a glass/ITO substrate; the corresponding molecular formula is described in Figure 1b. Shown in Figure 1c are the UV–vis absorption spectra of P3HT/PMMA bilayer films on glass for comparison. Strong absorption in the visible range can be observed and the absorption intensity determined by the content of P3HT in blend films. Under light illumination, the P3HT photosensitive layer excites electron–hole pairs and the holes are enriched in the channel to increase the drain current of the device. An absorption spectrum was also used as an approach to explore the inner thin film structure on this basis of a weakly coupled H-aggregate model [33,34]. The presence of a broad π–π* electronic transition band was proved by the peak absorption at 565 nm, and other two additional vibronic structures at 530 and 610 nm, which were related to coupling of C==C double bond symmetric stretch and the π–π* electronic transition. Further, the absorbance peak variations at 610 nm reflect the ordered aggregates formed via π–π stacking between polymer chains in three contents, facilitating transistor channel formation and carrier transfer.

To verify the phenomenon of vertical phase separation about PMMA and P3HT blend films, Figure 2 presents the results of surface contact angle we measured. The contact angles of pure PMMA and P3HT were 70° and 98°, respectively. Blend films spin coated by 1 mg/mL, 2 mg/mL, and 5 mg/mL P3HT solution shared the contact angle of about 97°. The performance was similar to pristine P3HT film, indicating that PMMA and P3HT had an obvious surface-induced vertical phase separation. Through a simple one-step spin coating process, a self-stratified bilayer structure of PMMA-bottom and P3HT-top would be naturally generated.

### 3.2. Phototransistor Performance

Figure 3 depicts the typical electrical characteristics of a P3HT/PMMA blend film device with different solution concentrations. There was no obvious hysteresis phenomenon observed from the transfer curves, showing that the interface between the dielectric-semiconductor layer prepared by the one-step method was more beneficial to carrier transport. Field effect mobility is the most important parameter for evaluating transistors. The specific numerical calculation in the saturation regime is given by the following equation:(1)$$I_{\mathrm{DS}}=\left(\frac{W}{2L}\right)\mu C_{i}{\left(V_{\mathrm{GS}}-V_{T}\right)}^{2}$$
where *L* and *W* represent the channel length and width of transistors, respectively. *C*_i_ is the capacitance (per unit area) of the dielectric layer, which was calculated by the two-parallel-plate model (Appendix A). The corresponding carrier mobility was 0.016 (cm^2^/Vs) of 1 mg/mL, 0.012 (cm^2^/Vs) of 2 mg/mL, and 0.007 (cm^2^/Vs) of 5 mg/mL. It is noteworthy that the devices also exhibited good field effect characteristics with low threshold voltage ranging from −3 to −1 V, and well-defined linear and saturation regimes; all the above parameters lay a foundation for the realization of phototransistors.

The transfer characteristics at different concentrations in darkness and under light illumination (1 mW/cm^2^, 8 mW/cm^2^) varied with V_GS_, as shown in Figure 4. Under the process of illumination intensity from dark state to 8 mW/cm^2^, it could be clearly observed that the photocurrent increased gradually and the threshold voltage showed a positive shift. The incident light induced non-equilibrium carriers in the P3HT semiconductor layer. Under the regulation of gate voltage, the free carrier generated moved to the channel at the interface of the insulator–semiconductor, where part of the carrier was used to increase the channel carrier density, and the other part was captured by the energy level defects of the insulating layer [15]. Furthermore, the dominant photovoltaic mode drives the electrons and holes in the conductive channel moving to the source and drain electrodes, which lead to band bending [13]. Compared with the transfer characteristics in darkness, a lower injection barrier makes a positive shift of the threshold voltage realized. Thus, the photocurrent if amplified and a positive shift of the threshold voltage occurs as the intensity of light increases.

The photoresponsivity and detectivity are two other essential parameters for evaluating the phototransistor performance. The responsivity as a function of V_GS_ under different P3HT contents with V_DS_ fixed at −40 V was presented in Figure 5a,b. Maximum responsivity was achieved at 597.6 mA/W under 1 mW/cm^2^ light intensity and 1 mg/mL content of P3HT. Further, 2 mg/mL and 5 mg/mL contents could also realize responsivity of 435.6 mA/W and 382.4 mA/W, respectively. By comparing the influence of 1 mW/cm^2^ and 8 mW/cm^2^ light intensities on the responsivity in detail, we obtained similar results to previous reports [13,14,15]; the value of responsivity decreased with the increase of light intensity, which indicated that the phototransistor fabricated by one-step spin-coating using the vertical phase separation method had good light detection performance. Figure 5d–f show the relationship between the detectivity of devices with gate voltage and light intensity, and it is clearly seen that the detectivity of the device reached the relatively large value of about 3.39 × 10^10^ Jones. At the same gate voltage, low light intensity achieved a higher detectivity. Further, photoresponsivity and detectivity as a function of different illumination powers are presented in Figure 5c,g; a certain degree of linearity is advantageous to the detection of different light intensities. A device with equivalent performance of 1 mg/mL P3HT content was further fabricated and studied. Figure 6 shows the results of the responsivity and detectivity with drain-source voltage under different gate voltages. It was revealed that the output signals of the phototransistors could be amplified by changing the drain-source or gate voltages. Specifically, the detectivity could achieve a maximum value of 4.25 × 10^10^ Jones under the condition of low drain-source and gate voltage. The performance improvement could be attributed to the small noise level of the current. Therefore, the phototransistor realized via phase separation of P3HT/PMMA blend film for weak light signal detection must be a very good choice.

### 3.3. Exploration of Microscopic Mechanism

Interestingly, all the phototransistors fabricated with three different solutions revealed the same result that 1 mg/mL P3HT and 100 mg/mL PMMA blend system realized the highest device performance, including photosensitivity, responsivity, and detectivity. To some extent, the degree of phase separation, crystallinity, conjugation length, macro-orientation of chains, and surface morphology in blend films have important effects on the performance of phototransistors.

To study its intrinsic micro-mechanism, Figure 7a–c presents the morphologies of the self-stratified films with different contents of P3HT. All of them showed rugged island-like features and an increase of roughness with root-mean-square (RMS) of 1.40 nm for 1 mg/mL film, 3.38 nm for the 2 mg/mL film, and 5.39 nm for the 5 mg/mL film. As RMS increased, there existed more defects on the surface to capture carriers, which would be against the carrier transport and have a negative impact on the light–dark current [35]. Furthermore, the SEM topography images of different contents are also shown in Appendix A. The results also explained well the correlation changes with the electrical characteristic of OTFTs. TEM was conducted for 1 mg/mL P3HT blend films to investigate the phase separation. As shown in Figure 7d, P3HT and PMMA regions can be clearly distinguished, especially through the chain formation characteristics and high density of P3HT. Dark regions and bright regions were described as P3HT-rich and PMMA-rich, respectively. The blend film had a good homogenization in the mass, among which PMMA presented an amorphous state and P3HT crystallized to form a continuous conjugate long chain and domains. The corresponding carrier mobility was 0.016 (cm^2^/Vs), which was better than other concentrations. Under light illumination, photogenerated carriers transported into the interface where vertical phase separation occurred between P3HT/PMMA; the high carrier mobility with a concentration of 1 mg/mL would naturally promote the directional flow of carriers in the channel and realize larger values of the photocurrent.

Generally, the final vertical phase separated structures of PMMA-bottom and P3HT-top formed by the simple one-step method depend on many factor, including solvent evaporation time, spin coated speed, and the solubility of conjugated polymers. When the blend of P3HT and PMMA was spin-coated onto a glass/ITO substrate, the current equilibrium would naturally be broken and P3HT with low surface energy was shown to migrate to the film/air interface, while the PMMA component with higher affinity for substrates preferentially migrated to the substrate interface [25,27]. The above phenomenon was the inevitable result of reaching thermodynamic equilibrium and minimizing the free energy of the system. However, polymers with low solubility in solvents have a disposition to aggregate during rapid solvent evaporation. According to nucleation and growth mechanisms, the liquid–liquid demixing process was inhibited by the occurrence of the dominant solid–liquid demixing, phase separation interrupted in the original direction, the resulting film morphology was affected by abundant aggregation, and very few components of the blend remained unseparated in a vertical composition gradient [27,36,37]. In our experiment, phototransistors with significant differences in performance were realized by adjusting the content of P3HT conjugated polymers. Here, 1 mg/mL of the P3HT solution could be rapidly dissolved in chlorobenzene at room temperature, but 2 mg/mL and 5 mg/mL solutions were completely dissolved at 80°, which showed that the solubility of P3HT polymers we used was poor in the corresponding chlorobenzene solvent at room temperature. With the P3HT content increased to 2 mg/mL and 5 mg/mL, low solubility resulted in the film morphology deterioration due to its abundant aggregation at the air/film interface. As shown in the previous AFM images, severe aggregation of 5 mg/mL P3HT can be seen from the almost isolated island morphology, which has certain limitations on vertical phase separation. On the contrary, even though there existed a slight aggregation phenomenon in 1 mg/mL, they uniformly dispersed and had the smallest RMS value at the air/film interface; vertical phase separation worked well during the rapid evaporation. Then, the disordered PMMA/P3HT coexistence region that was limited by aggregation of P3HT would affect the dielectric insulation of devices [36], including the formation of conductive channel and carrier transport. Consequently, the results well supported the correlation analysis about the effect of solubility on phase separation. Combining all of the above mechanisms, it is reasonable that the P3HT with 1 mg/mL device fabrication could achieve a higher performance phototransistor.

## 4. Conclusions

In conclusion, a simple one-step spin coated method for fabricating phototransistors was realized via vertical phase separation with a P3HT/PMMA blend. Owing to the ability of capturing and converting incident light into detectable electrical signals, P3HT phototransistors with different concentration have been extensively investigated. The device with 1 mg/mL P3HT delivers photodetection performance in terms of the photoresponsivity of 597.6 mA/W and detectivity of 4.25 × 10^10^ Jones under 1 mW/cm^2^ light intensity. All the above values revealed that continuously adjusting the content of P3HT and exploring the optimum solvent evaporation conditions to achieve the best vertical phase separation are critical for high detection performance. To some extent, this simple one-step fabrication method could be used as a general strategy for developing high performance phototransistor in the blend system with the channel materials that have both high carrier mobility and efficient light absorption capability.

## Figures and Tables

**Figure 1 micromachines-10-00716-f001:**
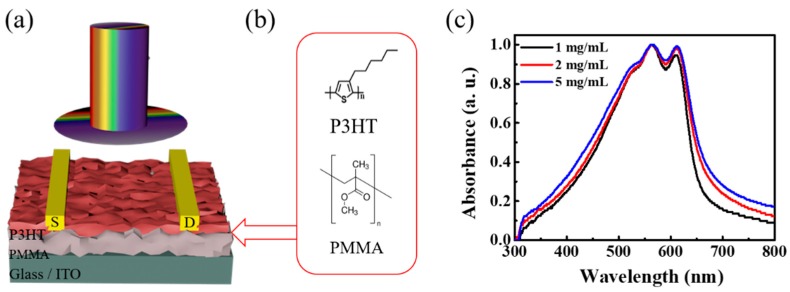
(**a**) Schematic diagram of the phototransistor. (**b**) Molecular formulas for poly(3-hexylthiophene-2,5-diyl) (P3HT) and poly(methyl methacrylate) (PMMA). (**c**) Absorption spectra of the blend films with different P3HT contents.

**Figure 2 micromachines-10-00716-f002:**
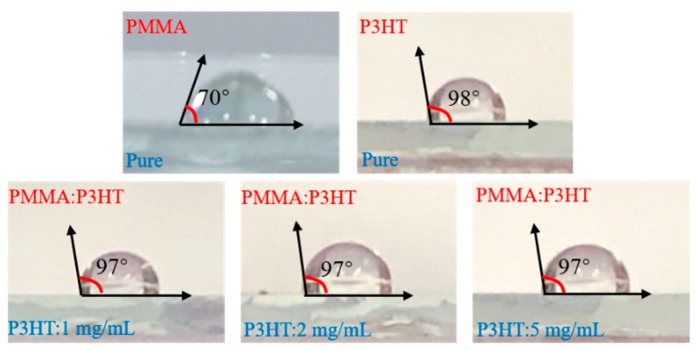
The contact angle of surface in pure PMMA, P3HT, and blend films with different contents of P3HT.

**Figure 3 micromachines-10-00716-f003:**
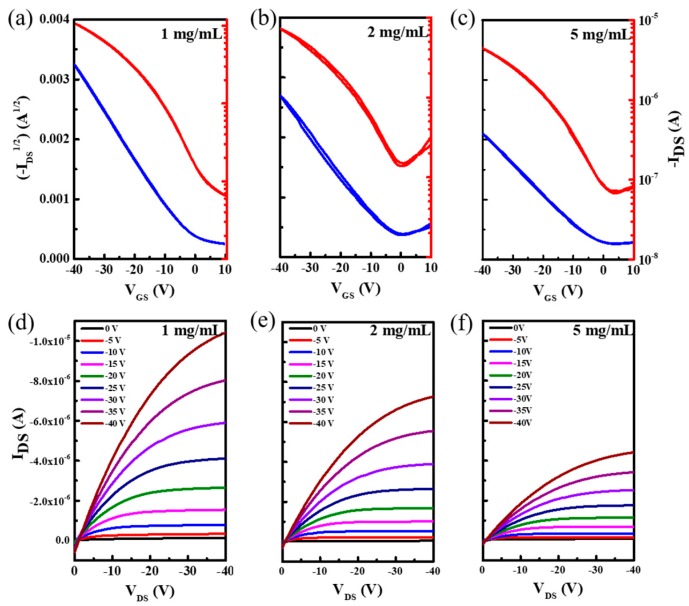
Electrical characteristic of organic thin-film transistors (OTFTs). (**a**–**c**) Transfer characteristics of P3HT OTFTs via different concentrations. (**d**–**f**) Output characteristics of OTFTs via different P3HT concentrations (the concentrations represented from left to right are 1 mg/mL, 2 mg/mL, and 5 mg/mL, respectively).

**Figure 4 micromachines-10-00716-f004:**
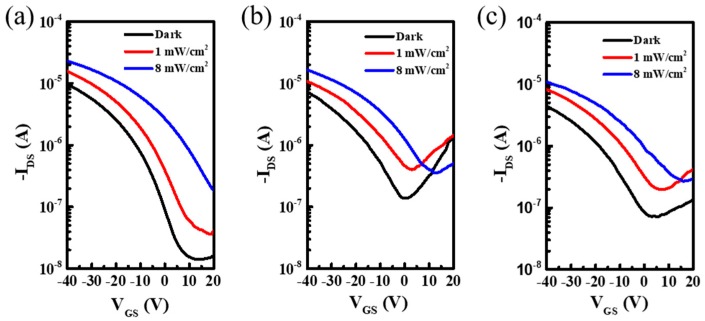
Transfer characteristics at different P3HT concentrations in darkness and under light illumination (1 mW/cm^2^, 8 mW/cm^2^) vary with V_GS_; the drain-source voltage was controlled at −40 V. (**a**) 1 mg/mL, (**b**) 2 mg/mL, (**c**) 5 mg/mL.

**Figure 5 micromachines-10-00716-f005:**
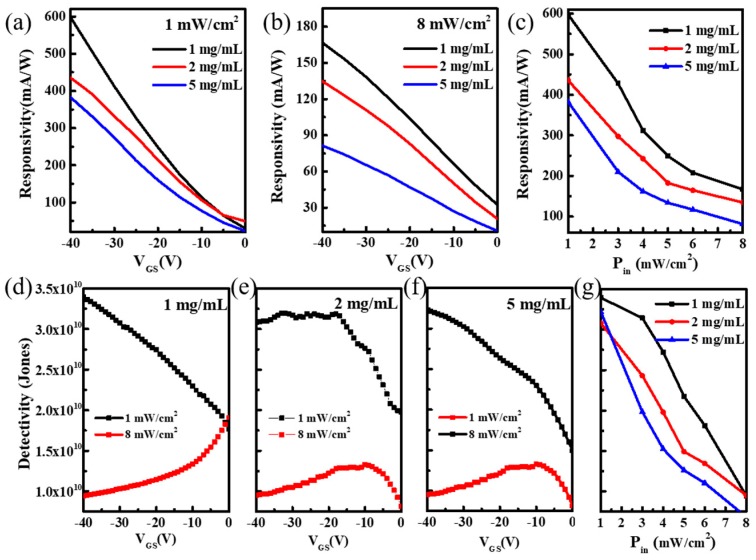
(**a**,**b**) Device responsivity as a function of the gate voltage under light illumination at intensities of 1 mW/cm^2^ and 8 mW/cm^2^, respectively. (**c**) Device responsivity as a function of different illumination. (**d**–**f**) Device detectivity as a function of the gate voltage under different P3HT concentrations of 1 mg/mL, 2 mg/mL, and 5 mg/mL, respectively. (**g**) Device detectivity as a function of different illuminations.

**Figure 6 micromachines-10-00716-f006:**
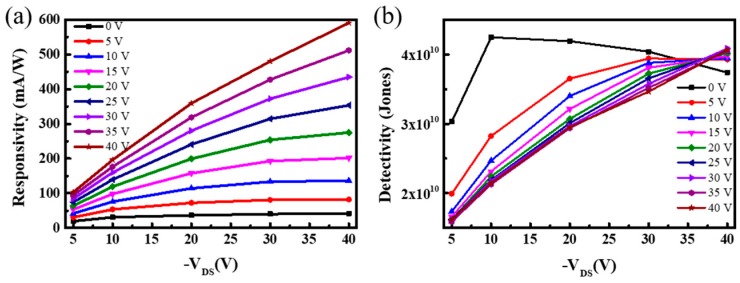
(**a**) Variation of device responsivity with drain-source voltage under different gate voltages. (**b**) Variation of device detectivity with drain-source voltage under different gate voltages.

**Figure 7 micromachines-10-00716-f007:**
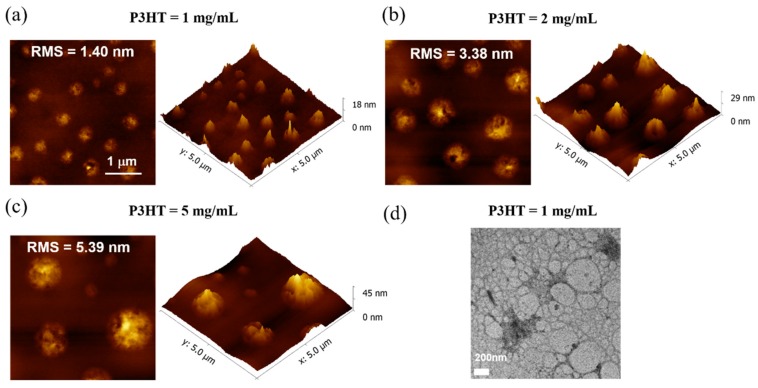
(**a**–**c**) Atom force microscopy (AFM) images of the P3HT/PMMA blend films with different contents of P3HT. (**d**) TEM images of the P3HT/PMMA blend films with 1 mg/mL P3HT.

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
