# Peer review of "One-Step Coating Processed Phototransistors Enabled by Phase Separation of Semiconductor and Dielectric Blend Film"

_micromachines, 2019, doi:10.3390/mi10110716_

Round 1
Reviewer 1 Report
This is an interesting study with solid and sound experimental results. I only have one minor question about the photoresponse characterization. For phototransistors, one of my concerns is the nonlinearity. I'd like to suggest the authors include two plots of the responsivity and detectivity as a function of light intensity.
Reviewer 2 Report
The authors have utilized a well-known technique of vertical phase-separation of polymer layers during formation of thin-films; they used this technique to make phototransistors based on light absorption in P3HT. They demonstrated some relation between P3HT concentration and photoresponse while reporting remarkably high D* for the resulting devices.
The vertical phase-separation employed is not new and early work in this area used precisely the same materials for semiconductor (P3HT) and dielectric (PMMA) as the authors use here. The originality and significance of the work is low because the authors do not provide any new insight into novel devices or materials, but rather fill-in-the-blank between 1) P3HT/PMMA FETs based on vertical phase-separation, 2) P3HT-based phototransistors, and 3) P3HT photodetectors/photovoltaics. These last two have been studied extensively and the photoresponse of P3HT is well known - demonstrating photoresponse of the first one seems relatively insignificant. However, if these devices make exceptional photodetectors, it would certainly be an impactful publication.
It is not clear that these make exceptional photodetectors, despite the large detectivity reported. The authors need to rigorously justify their detectivity results with detailed description of the experimental characterization and measurement of D*. In particular show noise measurements at the different bias conditions that you cite D*. Second, the reported responsivity seems ~ 10x too large; considering 8mW/cm^2 illumination in 4(a) against the dark looks like ~10 microA of photocurrent produced by 0.008*0.1 (irradiance*device area). Perhaps the device is not uniformly illuminated over the full area and the authors likely accounted for this, but it is not clearly detailed in the experimental methods. Their methods should detail the illumination setup. If the top hat in Figure 1(a) is illustrating something about the illumination, it is confusing and unclear.
To support the case for exceptional photodetectors from this phototransistor architecture, the transfer curves should represent state-of-the-art. However, these are relatively poor devices with low ON/OFF ratios, large OFF currents and require large bias due to the long channel length. The work would be more compelling with small sizes and low bias voltages useful for photodetectors, and this materials system should be able to better than reported here. The OFF currents in devices with higher P3HT content are particularly bad and may provide key insight into understanding the relative behavior of these devices.
The authors provide very limited insight into the important materials and device phenomena that drive the characteristics of these devices. More consequential microscopy, spectroscopy and/or modelling is needed to explain the phase-segregation, aggregation and, ultimately, charge-transport pathways in these devices. The cartoon in Fig. 7 is an inadequate representation of morphology while the authors don't provide sufficient evidence to support conclusions on stratification and transport. The contact angle measurements provide nice, but indirect, evidence that they form complete P3HT films on top of PMMA. But how thick are those neat P3HT layers? Are they nice and continuous to support charge transport? Are they truly pure, without residual PMMA? What drives the microscale aggregation that is observed? Some cross-sectional EDAX or XPS-imaging to show Sulfur distribution would provide definitive, chemically-selective data on the phase-segregation and composition distribution.
